# Ultrafast long-range spin-funneling in solution-processed Ruddlesden–Popper halide perovskites

David Giovanni [1,2], Jia Wei Melvin Lim [1,3], Zhongcheng Yuan [4], Swee Sien Lim [1], Marcello Righetto[1], Jian Qing[4], Qiannan Zhang [1], Herlina Arianita Dewi[2], Feng Gao [4], Subodh Gautam Mhaisalkar [2,5], Nripan Mathews[2,5] & Tze Chien Sum [1]

Room-temperature spin-based electronics is the vision of spintronics. Presently, there are few suitable material systems. Herein, we reveal that solution-processed mixed-phase Ruddlesden–Popper perovskite thin-films transcend the challenges of phonon momentum-scattering that limits spin-transfer in conventional semiconductors. This highly disordered system exhibits a remarkable efficient ultrafast funneling of photoexcited spin-polarized excitons from two-dimensional (2D) to three-dimensional (3D) phases at room temperature. We attribute this efficient exciton relaxation pathway towards the lower energy states to originate from the energy transfer mediated by intermediate states. This process bypasses the omnipresent phonon momentum-scattering in typical semiconductors with stringent band dispersion, which causes the loss of spin information during thermalization. Film engineering using graded 2D/3D perovskites allows unidirectional out-of-plane spin-funneling over a thickness of ~600 nm. Our findings reveal an intriguing family of solution-processed perovskites with extraordinary spin-preserving energy transport properties that could reinvigorate the concepts of spin-information transfer.

[1] Division of Physics and Applied Physics, School of Physical and Mathematical Sciences, Nanyang Technological University, 21 Nanyang Link, Singapore 637371, Singapore. [2] Energy Research Institute @NTU (ERI@N), Research Techno Plaza, X-Frontier Block, Level 5, 50 Nanyang Drive, Singapore 637553, Singapore. [3] ERI@N, Interdisciplinary Graduate School, Nanyang Technological University, 50 Nanyang Avenue, S2-B3a-01, Singapore 639798, Singapore. [4] Department of Physics Chemistry and Biology (IFM), Linköping University, 581 83 Linköping, Sweden. [5] School of Materials Science and Engineering, Nanyang Technological University, Nanyang Avenue, Singapore 639798, Singapore. Correspondence and requests for materials should be addressed to T.C.S. (email: Tzechien@ntu.edu.sg)

The future of spintronics hinges on semiconductors that enable efficient room temperature spin generation and transport[1–4]. Significant research efforts are devoted to search for a suitable candidate. Forerunner research on inorganic semiconductors (e.g., GaAs[1–4], MnSe[5], Si[6], etc.) face not only cost and integration challenges from high vacuum and temperature processing but also lattice-matching requirements. Organic semiconductors offer the advantages of facile fabrication and versatility[7–9]. While the weak spin-orbit coupling (SOC) in organics would theoretically enable long-range transport of spin current[7,8], it also regrettably imposes huge challenges for spin control[8,9]. Spin relaxation via hyperfine interaction in organics also limits its spin-relaxation lifetimes[7–9]. Recently, spintronic studies on low SOC inorganic system, such as graphene[10] and Si[11], have yielded exciting results, where room temperature spin-transport length of ~ 30 μm and ~ 20 μm, respectively, have been demonstrated. However, they are also limited by several fundamental issues: both Si and graphene have a very low SOC, in which, while it grants long spin diffusion, it also fundamentally limits the viability for spin control (optically and electrically). Moreover, these two systems require good quality single-crystal samples with minimal scattering centers to achieve long-range spin drift, thus necessitates costly fabrication techniques. Lastly, lattice mismatch and unintentional doping due to substrate effect are also major problems in Si and graphene, respectively.

Amidst this impasse, organic–inorganic lead halide perovskites emerge as prospective spintronic materials[12–16]. The defect tolerance and solution processability of lead halide perovskites may offer advantages over other material systems. Their huge SOC modifies their band edge into total angular-momentum states, or J-states of $|m_J\rangle = \pm 1/2$[12–17], thereby permitting optical excitation of J- (spin-) polarized carriers/excitons with 100% spin polarization [see Supplementary Note 2]. Nevertheless, this also restricts the spin lifetime to only a few ps in room temperature[14] and up to ~1 ns in cryogenic temperature[15,16]. This constraint would limit effective spin transport and could potentially derail halide perovskite's spintronics aspirations.

Most recently, moisture resilient mixed-dimensional lead halide perovskites, also known as Ruddlesden–Popper perovskites (RPP) (i.e., $A_2B_{n-1}Pb_nX_{3n+1}$; where A and B are long-chain and small organic cations, respectively; and X = Cl, Br, or I is the halide anion) made their debut in photovoltaics[18] and light emitting diodes (LED)[19]. RPP are perovskites with structural dimensionality in between layered 2D and infinite 3D perovskites (Fig. 1a). The crystal structure of RPP forms a natural multiple quantum-well (QW) system with alternating layers of long-chain organic cations [e.g., $C_6H_5C_2H_4NH_3^+$ as the barrier] and inorganic $[PbX_6]^{4-}$ octahedrons as the QW, with well thickness increasing proportionally with the phase number $n$. The bandgaps of RPP monotonically red-shift as $n$ increases from 2D ($n = 1$) to 3D ($n = \infty$). Recently, efficient ultrafast energy funneling (<0.5 ps) via quantum coupling from small $n$ toward large $n$ phases was demonstrated in a mixed phase RPP[19,20], with efficiency exceeding 85%[20]. Such coupling inspired us to hypothesize that spin funneling should also be feasible in the RPP system.

Herein, our findings reveal that the photoexcited spin-polarized excitons in low-$n$ funnel to high-$n$ via intermediate states (IS)-assisted energy transfer between two adjacent phases/domains. This process overcomes the scattering-induced spin relaxation typically found in conventional systems, permitting spin polarizations to be preserved during the transfer between the phases/domains. Unidirectional spin funneling in 2D/3D layered thin films over its 600 ± 80 nm thickness is also demonstrated. This novel approach of spin funneling could reinvigorate the concepts of transporting spin information using halide perovskites.

## Results

**The concept of spin funneling.** In this study, two standard RPP thin film samples with two different precursor stoichiometric ratios are used: $\bar{n} = 2$ and $\bar{n} = 4$. Note the two indices: $\bar{n}$, which refers to the precursor solution stoichiometric ratio according to the chemical formula $(C_6H_5C_2H_4NH_3)_2(CH_3NH_3)_{n-1}Pb_nI_{3n+1}$; and $n$, which refers to the RPP phase index. These two $\bar{n} = 2$ and $\bar{n} = 4$ samples form mixtures of RPP phases of $n = 1, 2, 3, …, \infty$, with different distributions, as can be seen from their absorption and photoluminescence (PL) spectra (Fig. 1b). The total angular momentum ($J$-) or spin-selection rule for circularly polarized light is shown in Fig. 1c. Excitation by circularly polarized light $\sigma^\pm$ will create hot $J$- (hereafter called as spin-) polarized carriers/excitons, which will thermalize toward the band edge.

Based on this, we envisage the concept for efficient spin funneling as presented in Fig. 2a, b. For the case of nonresonant photoexcitation, as the hot carriers/excitons thermalize toward the band edges, the stringent parabolic energy-momentum dispersion in conventional semiconductor requires a momentum-scattering process to occur (Fig. 2a), as observed in pure the 3D perovskites ($n = \infty$ or $CH_3NH_3PbI_3$—henceforth MAPI). Such scatterings [i.e., with carriers, impurities, or longitudinal optical phonons] inevitably cause the loss of spin information, typically via Elliot−Yafet mechanism at a rate proportional to the momentum-scattering rate[1–3]. This process of thermalization (i.e., momentum relaxation by phonons) to a certain extent would be similar with phonon dominated momentum-scattering processes, which causes relaxation of spin current during charge drifting-based spin transport in room temperature[3]. However, in mixed RPP (Fig. 2b), the presence of IS between two $n$ phases bypasses the momentum-scattering process, and the photoexcited excitons funnel down the RPP states via IS-assisted energy transfer.

Our hypothesis is proven by circular-polarized transient absorption (TA) spectroscopy (or known as pump probe, see 'Methods" for details). Herein, the change of probe transmission ($\Delta T/T$) induced by pump pulse could be used to measure the population of the exciton spin states, hence allowing a time-resolved monitoring of the spin dynamics. Figure 2c shows the initial ($t = 0.6$ ps) spin polarization of the MAPI thin film at its band edge (~1.63 eV) photoexcited by the circularly polarized pump. The spin polarization is defined as the difference between co-circular and counter-circular spin-polarized TA signal divided by their total:

$$P = \frac{(\Delta T/T)_{\text{co}} - (\Delta T/T)_{\text{counter}}}{(\Delta T/T)_{\text{co}} + (\Delta T/T)_{\text{counter}}}. \quad (1)$$

Note that the polarization calculated here is the system averaged spin polarization of the population, comprising of a mixture of excitons and free carriers. As illustrated in Fig. 2a, MAPI stringent band dispersion limits the available carriers' phase space, and hence momentum scattering is inevitable. Assuming the spin-flip probability due to momentum scattering during thermalization to be $\alpha$, the observable spin polarization is given by:

$$P = P_0 e^{-\alpha \Delta E}. \quad (2)$$

Here, $P_0$ is the photoexcited spin polarization and $\Delta E = E_{\text{pump}} - E_{\text{probe}}$ is the energy difference between the pumping and the probing state (band edge at 1.63 eV), i.e., energy loss due to thermalization. Here, Eq. (2) satisfactorily describes our observations in MAPI (Fig. 2c).

However, contrasting behavior is observed in the initial spin polarization ($t = 0.6$ ps) of the RPP samples ($\bar{n} = 4$) photoexcited by the 2.07 eV circular pump (Fig. 2e). A proof of spin polarization in

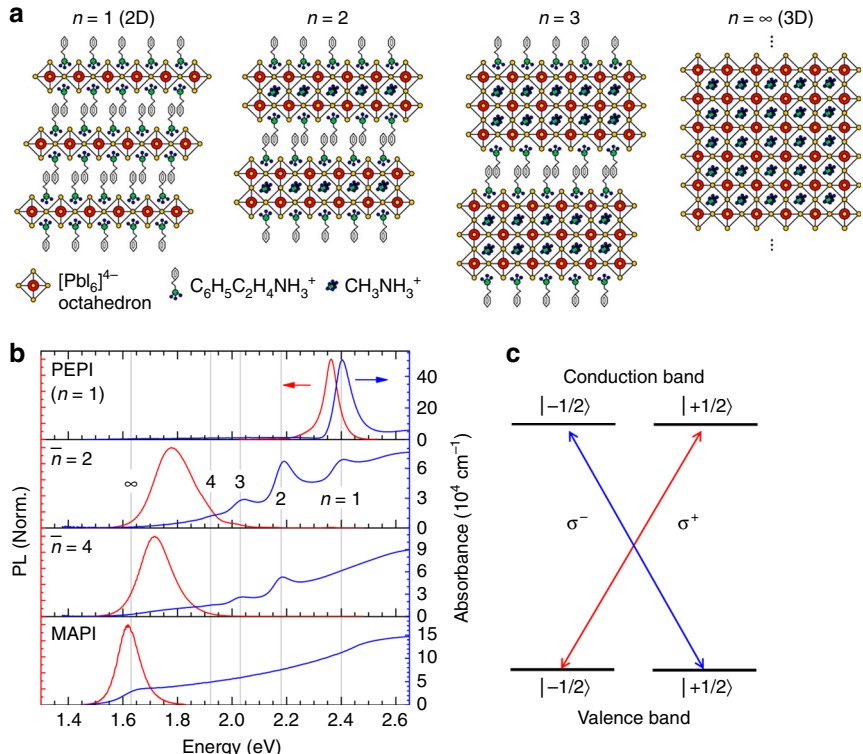

**Fig. 1** Mixed phase Ruddlesden–Popper perovskite (RPP). **a** Crystal structure of several phases of lead iodide RPP: $n = 1$ (pure two-dimensional), $n = 2$, $n = 3$, and $n = \infty$ (pure three-dimensional). The crystal structure goes with formula $(C_6H_5C_2H_4NH_3)_2(CH_3NH_3)_{n-1}Pb_nI_{3n+1}$. **b** Absorbance and photoluminescence (PL) spectra of our RPP samples for different stoichiometric ratio of the precursor solution: $\bar{n} = 1$ (PEPI), $\bar{n} = 2$, $\bar{n} = 4$, and $\bar{n} = \infty$ (pure 3D or MAPI). For $\bar{n} = 2$ and $\bar{n} = 4$ samples, mixed multiple phases of RPP are observed. Here $n$ refers to the actual phase of the RPP, while $\bar{n}$ refers to the stoichiometric ratio of the precursor solutions. **c** Selection rule of circularly polarized light based on the total angular momentum in perovskites' band edge with total angular momentum states $|m_J\rangle = \pm 1/2$

our RPP samples is shown through a modified Hanle measurement (see Supplementary Note 11). An initial growth followed by slight decrease and then back to an increasing spin-polarization trend is observed with increasing $\Delta E$ (average of $0.46 \pm 0.05$); and with higher spin polarization than in MAPI at similar $\Delta E$ (i.e., ~1.8× at $\Delta E = 0.3$ eV). This different trend and higher spin polarization are sufficient to imply that the spin polarization at RPP $n \geq 4$ phases (lower bandgap) originates dominantly from the sequential transfer of lower $n$ phases (higher bandgap), rather than from thermalization of photoexcited hot carriers/excitons within itself.

This trend itself can be semi-quantitatively modeled by a combination of trap-limited funneling of spin-polarized excitons down towards the lower bandgap phases/domains with a minor contribution from the thermalization process (vide infra, Supplementary Note 5). An $F$ test on this model shows 95% significance ($F = 6.14$ with $p = 6$, against the mean), which confirms the validity of the model. Qualitatively, this trend can be described as the following. The initial growth followed by a decrease at $\Delta E < 0.2$ eV arises from the interplay between the spin-funneling process from $n = 3$ to $n = 4$–$5$ phases, and the thermalization of the exciton within $n = 4$–$5$ phases itself. It is then followed by a plateau at $0.2$ eV $< \Delta E < 0.4$ eV, which signifies spin polarization that predominantly originates from the spin funneling; and the final increase arises from density-dependent diffusion/funneling due to traps. Further proof of funneling can be observed from the prominent increase of signal rise time with increasing $\Delta E$ in our $\bar{n} = 4$ RPP samples (Fig. 2f, h), as compared to the faster thermalization process in our MAPI sample, which is weakly dependent on $\Delta E$ within our temporal resolution (Fig. 2d, g)—further suggesting that the spin polarization does not come from thermalization. We assigned this process

to originate from IS-mediated energy transfer process, which preserves the spin and results in higher initial polarization.

**TA study on IS**. To validate the presence of IS, we performed TA studies across the visible spectrum. Figure 3a, b shows the TA spectra of $\bar{n} = 2$ and $\bar{n} = 4$ samples, using 2.07 eV pump (10 μJ/cm²) at time delay $t = 0.6$ ps. There are two possible contributions to the TA signal: (1) positive $\Delta T/T$ contribution [photobleaching (PB)] arise from either state filling (SF) or stimulated emission (SE) by the probe pulse from a populated excited state; and (2) negative $\Delta T/T$ contribution [photoinduced absorption (PA)] arise from probe absorption from a populated excited state to higher energy states. We assigned the observed broad PA band at probe energy $\hbar\omega > 2.1$ eV to arise from absorption of the photo-excited excitons in lower bandgap phases to higher energy states. This above-bandgap PA band is commonly observed in per-ovskites[21], and is not the focus of this work. The PB signal, which spectrally coincides with the absorption peak/edge of a RPP phase, arises from the SF effect and corresponds to the none-quilibrium exciton population in that phase. For both samples, the $n = 3$ (~2.03 eV) and $n = 4$ (~1.92 eV) PB peaks were clearly observed. For $n > 4$ phases ($\hbar\omega < 1.9$ eV), the $\bar{n} = 2$ and $\bar{n} = 4$ samples show broad PB bands up to ~1.80 and ~1.68 eV, respectively, which correspond to a population of convolved $n \gg 1$ (3D-like) phases. The broader PB band in $\bar{n} = 4$ samples is due to the presence of more $n \gg 1$ phases than in $\bar{n} = 2$ samples, which is also consistent with their linear absorption spectra (Fig. 1b).

A weaker counter-circular pump-probe polarization PB signal than its co-circular signal indicates an imbalance of exciton

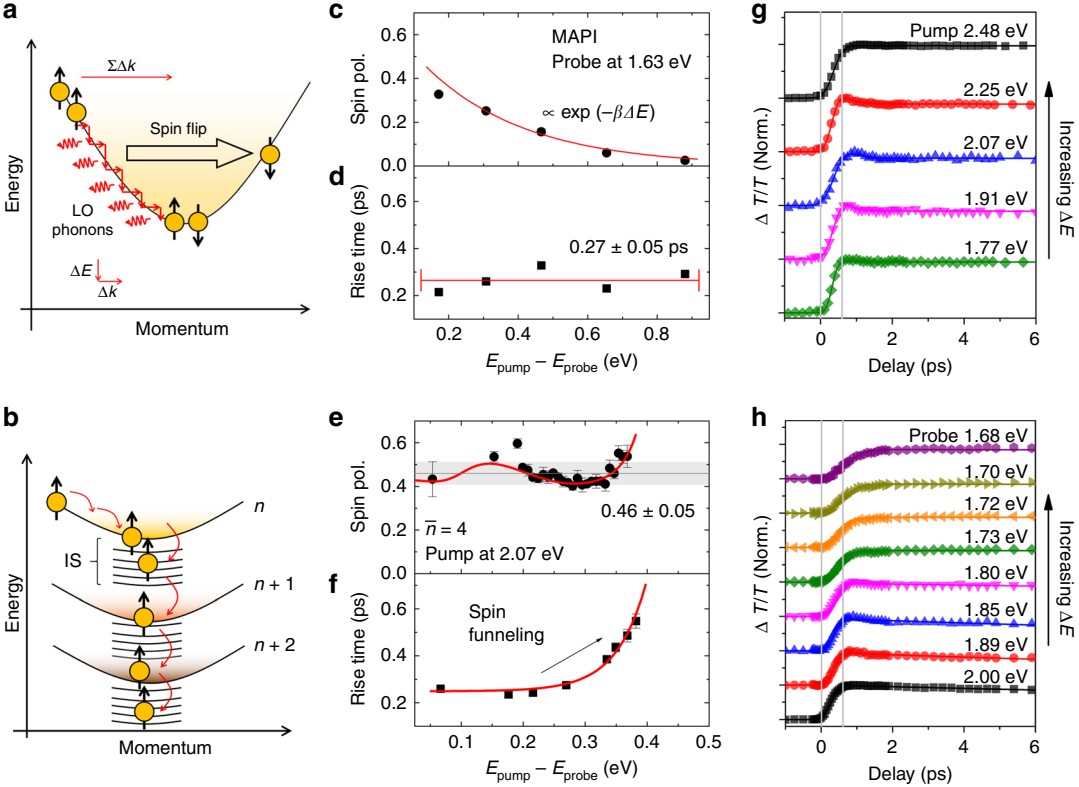

**Fig. 2** Spin funneling in Ruddlesden–Popper perovskite (RPP). **a** Band structure with stringent energy-momentum dispersion (such as in 3D $CH_3NH_3PbI_3$ or MAPI) requires hot spin-polarized carriers/excitons to undergo momentum scattering with longitudinal-optical (LO) phonons to thermalize to the band edge, causing spin flip via the Elliot–Yafet mechanism. **b** Band structures of our mixed-phase RPP. The spin-polarized carriers/excitons funnel down from low-$n$ to high-$n$ phases via intermediate states (IS), which bypasses the momentum scattering and preserves the spin polarization. The initial (pump-probe delay $t = 0.6$ ps) spin polarization and $\Delta T/T$ signal rise time as function of $\Delta E = E_{pump} - E_{probe}$ of **c**, **d** the MAPI thin film probed at its band edge of 1.63 eV; and of **e**, **f** RPP $\bar{n} = 4$ thin films pumped at 2.07 eV. The RPP sample shows average spin polarization of 0.46 ± 0.05. The spin-polarization data in **e** is fitted by our trap-limited energy diffusion model (see Supplementary Note 5), while the error bars are obtained from the standard deviation of spin polarization measured within the probe spectral bandwidth of 3 nm. Bleaching $\Delta T/T$ signal rise in **g** MAPI and **h** RPP $\bar{n} = 4$ thin films. Significant slowing of the signal rise time with increasing $\Delta E$ is observed for the latter. Vertical lines are eye guides at $t = 0$ and $t = 0.6$ ps

populations in different spin states (i.e., nonzero spin polarization), generated by our circularly polarized-pump beam. Details of the decay kinetics of these spin-polarized excitons are given in the Supplementary Notes 7 and 8. At this early time, no PB signal is observed at $n = 2$ level (~2.18 eV), implying that this phase is unpopulated. This is expected because of our below-bandgap pump energy. However, due to RPP's type-II interphase band alignment[19,22], slower charge transfer (CT) process would later populate the $n = 2$ phases in a time scale of few ps (Supplementary Note 3). Such CT process has been reported by Bouduban et al.[23] in lead bromide-based RPP.

Meanwhile, interpretation for the TA signal at $\hbar\omega$ in between two RPP phases can be nontrivial. Figure 3c, d shows the co- and counter-circular kinetics at $\hbar\omega = 1.98$ eV, in between of $n = 3$ and $n = 4$ phases. For both samples, the co- and counter-circular TA signal show different signs. While temporal switching from PA to PB or vice versa (i.e., change of sign) are commonly observed in inorganic nanostructures[24,25] and 3D perovskites[26] due to two different states/populations (different lifetimes) sharing the same spectral region, it is not the case here. In our case, there are three features to highlight: firstly, the PA and PB signals occurred concomitantly for different polarizations with identical dynamics, but different signs. This implies different optical transitions of the probe with different helicity, each one involving different spin polarization, originating from the same initial energy state. Secondly, the observation of PB signal ($\Delta T/T > 0$, co-circular) in

this interphase spectral region (between the band edges of $n = 3$ and $n = 4$) implies the presence of populated energy states in between these two well-separated bandgaps. Lastly, the $\Delta T/T$ signal peak intensities for co- and counter-circular system show linear and saturating trends with increasing pump fluence, respectively (Fig. 3e). The former implies that no multiparticle process is involved and the saturation of pump absorption has not been reached; while the latter shows the saturation characteristic of a nonradiative trap-filling process, with saturation fluence for $\bar{n} = 4$ films is 24 ± 2 μJ/cm². Also, note that there is a slight shift (approximately few meV) of the excitonic resonance peak between co- and counter- circular pump-probe signal. This shift does not affect our interpretation/calculation of the spin polarization (see Supplementary Note 6).

Without loss of generality, we assume initial excitation by $\sigma^+$ pump that generates a population of $|+1\rangle$ excitons. The positive $\sigma^+$ (co-circular) $\Delta T/T$ signal would then arise from the SF/SE of $\sigma^+$ probe by $|+1\rangle$ excitons; while the negative $\sigma^-$ (counter-circular) $\Delta T/T$ signal would arise from PA/further absorption of $\sigma^-$ probe by $|+1\rangle$ excitons to higher energy states. Here, it is straight forward to assign the PB peaks observed at the $n$ and $n + 1$ bandgaps to arise from SF and SE of the band edge. Nevertheless, for the spectral region in between $n$ and $n + 1$ bandgaps, based on our arguments, the most plausible explanation is the presence of interphase energy state, or IS with shallow traps, whose populations are thermally connected via Boltzmann

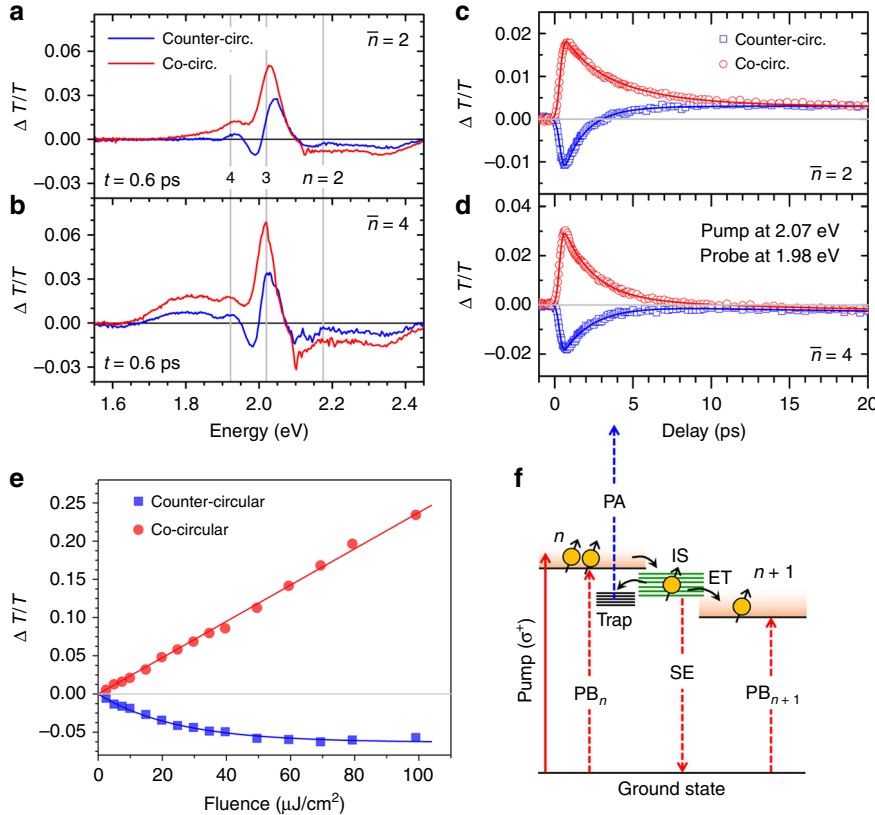

**Fig. 3** Assignment of intermediate states (IS) in Ruddlesden–Popper perovskite (RPP). Transient absorption spectra of **a** $\bar{n} = 2$ and **b** $\bar{n} = 4$ RPP thin film samples, pumped at 2.07 eV (pump fluence of 10 μJ/cm²) at pump-probe delay $t = 0.6$ ps. The corresponding kinetics at 1.98 eV (i.e., in between of $n = 3$ and $n = 4$ RPP phases) of **c** $\bar{n} = 2$ and **d** $\bar{n} = 4$ samples. The red and blue symbols correspond to co- and counter-circular pump-probe polarization, respectively. **e** Peak amplitude of co- and counter-circular signal on $\bar{n} = 4$ samples at 1.98 eV probe, showing linear and saturation trends with increasing pump fluence, respectively. **f** Assignment to IS (green) in between two adjacent RPP phases $n$ and $n + 1$. Assuming excitation of spin-up excitons by $\sigma^+$ pump, at an energy level between two adjacent RPP phases, the $\sigma^+$ and $\sigma^-$ probe signals originate from the transition via stimulated emission (SE, $\Delta T/T >$ 0) from the IS and via photoinduced absorption (PA, $\Delta T/T < 0$) from the trap states, respectively. PB refers to photobleaching, which is due to state filling at the RPP phases' band edges. ET refers to energy transfer. The pump and the probe are illustrated by the solid and dashed lines, respectively. The red and blue lines correspond to $\sigma^+$ and $\sigma^-$ polarization, respectively

distribution (i.e., hence sharing similar dynamics) (Fig. 3f). These IS could originate from either edge states[27], interwell two-exciton excited states[28], tail states, or grain boundaries. However, pinpointing the origin of these states is beyond the scope of the current work.

The PB signal by $\sigma^+$ probe would originate from SE of excitons from IS to ground state (i.e., $\Delta m_J = +1$), while the PA signal by $\sigma^-$ probe originates from further absorption of the trapped $|+1\rangle$ excitons in IS to $|0\rangle$ upper energy state (i.e., $\Delta m_J = -1$). This upper state is tentatively assigned to either the bound biexcitonic[29] state or the $|0\rangle$ $\Gamma_1^-$ state from $C$-band[17]. Filling of these traps would then cause saturation of the $\sigma^-$ (i.e., counter-circular) PA signal, as observed in Fig. 3e. As the spin polarization relaxes (toward an equal population of $|+1\rangle$ and $|-1\rangle$ excitons) in a timescale of few ps, the co- and counter-circular TA signal merges, as both SE and PA transitions from IS no longer discriminate exciton spins.

The remaining weak TA signal would then originate from the difference in transition strength between PA and SE from IS. This resultant signal is strongly dependent on the IS trap densities, which would differ based on sample types and preparation. It explains the difference of our observation of positive (SE > PA) and negative (SE < PA) post-spin relaxation TA signal for our $\bar{n} = 2$ and $\bar{n} = 4$ samples, respectively (Fig. 3c, d). Similar IS signatures are also observed at $\hbar\omega = 2.12$ eV [i.e., between $n = 2$

(~2.18 eV) and $n = 3$ (~2.03 eV) phases] when the samples were excited by 2.25 eV pump (Supplementary Note 4). This demonstration of IS hence supports our hypothesis of IS-mediated efficient spin funneling in RPP.

**Long-range spin transport via spin funneling.** Lastly, to demonstrate the viability of the spin-funneling approach to trans-ferring/transporting spin information for applications in opto-spin-tronics, optical spin generation, and directional spin funneling in RPP across 600 ± 80 nm thick graded films are presented. Figure 4a shows the schematic of our graded RPP film, where the RPP phases are engineered to gradually change from low-$n$ and high-$n$ phases from the bottom to the top of the film, forming a gradient 2D/3D per-ovskite stacking layer. Such a sample is designed to allow the pho-toexcited spin-polarized excitons to funnel unidirectionally from the bottom to the top of the sample. Layer stacking of the sample is validated with PL and reflectivity spectra from the front and back excitation of the sample (Supplementary Note 10). Figure 4b shows the TA spectra of our 2D/3D sample photoexcited by 2.07 eV pump (10 μJ/cm²) at $t = 0.6$ ps. Similar to the standard RPP $\bar{n} = 4$ samples (Fig. 3b), it shows PB peaks of $n = 3$ and $n = 4$ phases, but with a broader PB band spanning from ~1.63 to ~1.88 eV. This implies a better formation of $n \gg 1$ phases, in contrast with the standard $\bar{n} = 4$ samples. Stronger co- vs. counter-circular PB signal from 1.63 to

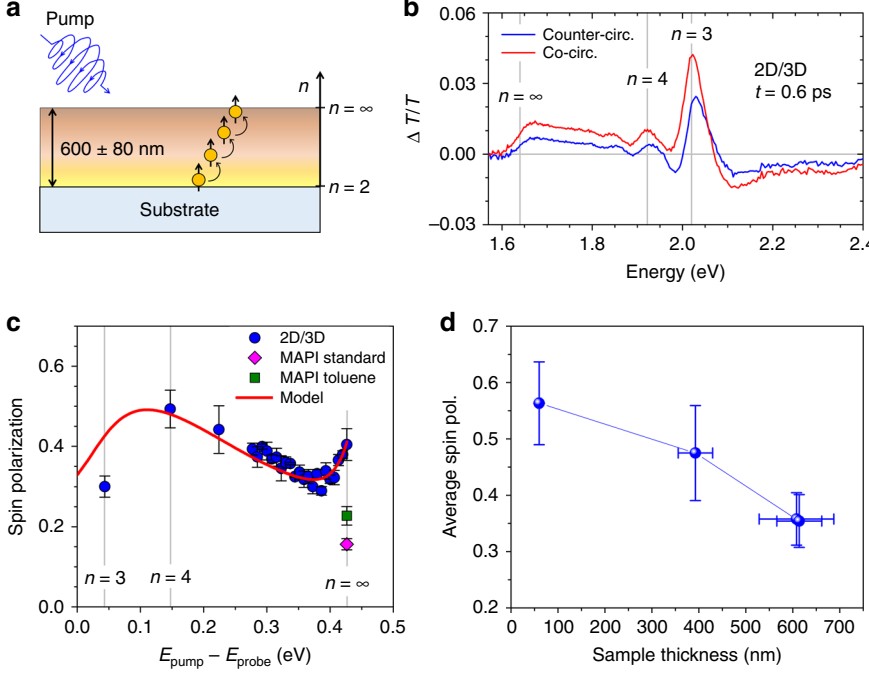

**Fig. 4** Directional spin funneling in Ruddlesden–Popper perovskite (RPP). **a** Schematic of the 2D/3D graded RPP samples. Excitation by $\sigma^+$ pump will generate spin-polarized excitons across the whole film thickness, followed by unidirectional funneling of the spin-polarized excitons from bottom (low-$n$) to top (high-$n$) phases of the film. No difference is observed when pumping from the front (air interface) or the back (substrate interface) sides of the samples. **b** Transient absorption spectra of the 2D/3D sample, pumped at 2.07 eV (pump fluence of 10 μJ/cm$^2$) from the front side, at pump-probe delay $t = 0.6$ ps. The red and blue spectra correspond to the co- and counter-circular pump-probe polarization. **c** Initial photoexcited spin polarization of our 2D/3D sample (blue circles) as a function of $\Delta E = E_{pump} - E_{probe}$, with average spin polarization 0.36 ± 0.05. The data is semi-quantitatively explained by trap-limited energy diffusion model (Supplementary Note 5). The early spin polarization (at $t = 0.6$ ps) of $CH_3NH_3PbI_3$ (MAPI) samples with standard (pink diamond) and toluene-washed spin-coating treatment (green square) are included for comparison. These spin polarizations originate from thermalization from the pump level to the band edge, showing lower polarizations as compared to the case of spin funneling in 2D/3D RPP samples. The error bars are obtained from the standard deviation of the spin polarization measured within the probe spectral bandwidth of 3 nm. **d** Thickness dependence of the average initial spin polarization (at $t = 0.6$ ps) of the 2D/3D graded samples, showing decreasing trend with increasing sample thickness. The error bars are obtained from the standard deviation of spin polarization across the spectra

2.07 eV spectral region indicates nonzero initial spin polarization of excitons populations in all $n \geq 3$ phases. Note that the interphase TA signal in this sample at $\hbar\omega = 1.98$ eV (between $n = 3$ and $n = 4$) also shows similar signatures of opposite signal signs for different polarization, as in the standard samples, implying that the same IS-assisted spin-funneling mechanism is also present.

Figure 4c shows the initial photoexcited exciton spin polarization of this 2D/3D sample as a function of probe energy—fitted with trap-limited energy diffusion model based on our findings in Fig. 3e (Supplementary Note 5). A similar trend, but slightly lower spin polarization than our standard $\bar{n} = 4$ samples, is observed with an average of 0.36 ± 0.05. A lower value is expected as the spin-polarized exciton needs to travel a further distance in this graded sample to reach the high-$n$ phases, in contrast to mesoscopic-like mixtures in the standard samples. For comparison, Fig. 4c also presents the spin polarization of MAPI ($n = \infty$) samples, prepared by standard spin-coating and anti-solvent (i.e., toluene) treatment techniques, at identical experimental conditions. While toluene treatment improves the post-thermalization spin polarization at the band edge (~1.64 eV) due to improvements in the morphology, it is by far inferior to the 2D/3D layered sample. The effect of 2D/3D sample thickness on the funneling is shown in Fig. 4d. The sample thickness measurement is shown in Supplementary Note 9. A decreasing trend of the average initial spin polarization with increasing thickness of the 2D/3D graded samples is observed. This result is expected as the spin information has to travel longer spatial distances.

## Conclusion

In summary, we demonstrated an efficient ultrafast funneling of photoexcited spin-polarized excitons at room temperature in RPP. Spin-polarized excitons funnel from the low $n$ (2D-like) to high $n$ (3D-like) phases mediated by IS-assisted energy transfer process, which bypasses the phonon momentum scattering during thermalization—thereby preserving the spin information. Unidirectional out-of-plane spin funneling in an engineered 2D/3D graded perovskite RPP film with thickness up to 600 ± 80 nm from the bottom (low $n$) to the top layer (high $n$) is demonstrated.

For comparison, the typical spin transport along a non-magnetic medium is through the charge-drifting[2–4,6,8,9] process. Such drifting process gives rise to a slow transport with plenty of collision events with phonons or defects, which compromise the spin polarization of the carriers as they travel to the counter electrodes for spin detection. This limits the distance of the room-temperature spin propagation length to only a few tens of nm in organics[9] and a few hundreds of nm in inorganic semiconductors[30–32]. Here, comparable transfer of spin information as for inorganic systems is achieved with perovskites, via this novel energy funneling mechanism.

The discovery of an efficient room-temperature spin funneling platform opens up exciting prospects for perovskite spintronics. We envision that future steps in this direction will focus on the spin control during the funneling process. As guidelines, we believe that either the control of the Fermi level via electric field

(E-field) and light or the control of the SOC strength in the material via E-field or magnetic field (B-field) are viable strategies for achieving effective spin control. Further studies can also include finding a suitable material for efficient injection and spin collection in perovskites. These studies, including this novel concept of spin funneling, could find applications as a new type of perovskite-based spin transistor, which is scattering and defect tolerant toward its spin transport properties. This exciting outlook derived from our current result paves the way for future research. Our findings highlight the significance of the spin-funneling process in RPP as a novel approach to transcend the strong SOC in perovskites to achieve efficient transfer/transport of spin-polarized excitons/carriers. This new family of halide perovskite materials with their extraordinary properties could provide a new spin on semiconductor spintronics.

## Methods

**Sample fabrication.** The MAPI precursor solution was prepared by dissolving 1:1 molar stoichiometric ratios of $CH_3NH_3I$ and $PbI_2$ powders into N,N-dimethyl-formamide (DMF) with 12.5 wt% concentration. The RPP precursor solutions were prepared by dissolving $C_6H_5C_2H_4NH_3I$, $CH_3NH_3I$, and $PbI_2$ powders into DMF with $2:\bar{n}-1:\bar{n}$ molar stoichiometric ratio with 0.25 M concentration. All films were prepared by spin coating the 20 μL of the precursor solutions on plasma-cleaned quartz substrates, followed by thermal annealing at 10 °C for 30 min. For anti-solvent treated samples, 90 μL of toluene was dropped on the substrate 3 s after the spin coating started. The 2D/3D graded sample was made by a precursor solution that was prepared by mixing $PbI_2$ (0.95 mol), $PbCl_2$ (0.05 mol), $CH_3NH_3I$ (0.85 mol), and $C_6H_5C_2H_4NH_3I$ (0.5 mol) in the mixture of N,N-DMF/dimethyl sulf-oxide (volume ratio 15:1). The concentration of $Pb^{2+}$ was maintained at 1 M. The mixture solution was stirred at room temperature for 6 h and filtered with a 0.2 μm filter before use. The precursor solution was spin coated on cleaned substrates (4000 rpm for 60 s) directly prior to 100 °C solvent annealing process, continued by films annealing for 10 min. All the preparation was conducted inside a dry $N_2$ glove box.

**TA spectroscopy.** The measurements were performed using a home-built chirp-corrected system with monochromator + photomultiplier tube + lock-in detection. It is powered by 800 nm Coherent Inc. Libra™ Ti:Sapphire laser (~50 fs, 1 kHz). The output was split into two beams: one was directed to an optical parametric amplifier (Coherent OPeRa SOLO™) to generation tunable photon energy (pump) and mechanically chopped at 83 Hz. The weaker beam (probe) was steered to a delay stage before white-light generation (1.5–2.8 eV) on a sapphire crystal. The measurement was performed in transmission mode. The pump and the probe were focused on the sample with $1/e^2$ spot diameter of ~600 and ~250 μm, respectively. Linear polarizers together with a Soleil–Babinet compensator and an achromatic quarter-wave plate were used to generate circular polarization for the pump and probe beams, respectively. See Supplementary Note 1 for details.

## Data availability

The data that support the findings of this study are deposited in the NTU open acess data repository DR-NTU: https://researchdata.ntu.edu.sg/dataset.xhtml?persistentId=doi:10.21979/N9/DUMOTD. The data are also available from the corresponding author upon reasonable request.

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

## Acknowledgements

Financial support from Nanyang Technological University start-up grants M4080514 and M4081293; the Ministry of Education AcRF Tier 1 grants RG104/16 and RG173/16 and Tier 2 grants MOE2015-T2-2-015, MOE2016-T2-1-034, and MOE2017-T2-1-001; the NTU-A*STAR Silicon Technologies Center of Excellence Program Grant 11235100003, from the US Office of Naval Research (ONRGNICOP-N62909-17-1-2155) and from the Singapore National Research Foundation (Program NRF-CRP14-2014-03, NRF2018-ITC001-001, and NRF-NRFI-2018-04) is gratefully acknowledged. F.G. acknowledges the financial support from the Joint NTU-LiU PhD programme on Materials and Nanoscience, the Swedish Government Strategic Research Area in Materials Science on Functional Materials at Linköping University (Faculty Grant SFO Mat LiU No 200900971), and the European Commission for the ERC Starting Grant (717026).

## Author contributions

T.C.S. and D.G. conceived the idea for the manuscript and designed the experiments. D.G., S.S.L., and J.W.M.L. conducted all the spectroscopic characterization. Q.Z., H.A.D.,

N.M., and S.G.M. prepared the standard RPP and MAPI samples. Z.Y., J.Q., and F.G. prepared the 2D/3D graded perovskite samples. Q.Z. performed the atomic force microscopy sample thickness measurement. T.C.S., M.R., and D.G. analyzed the data and wrote the manuscript. All authors discussed the results and commented on the manuscript at all stages. T.C.S. led the project.

## Additional information

**Competing interests:** The authors declare no competing interests.

**Peer Review Information:** *Nature Communications* thanks the anonymous reviewers for their contribution to the peer review of this work. Peer reviewer reports are available.

