## [Peer Review File · Nature Communications]

Reviewers' comments:

Reviewer #1 (Remarks to the Author):

It is my second to review this manuscript (it was submitted to Nature Photonics). As I stated earlier in my previous comment: "The discovery of such a long-time spin-dependent exciton would potentially introduce a novel concept of the transport of spin information in solution-processed hybrid perovskite thin films for quantum computing applications and communication technologies." It is noteworthy that these unique low-temperature solution-processed hybrid materials encompass rich spin-dependent phenomena (e.g., giant Rashba splitting, spin-selective optical Stark effect, spin control induced by chirality, etc.) that are designed to manipulate within the same material class, and they can be synthesized using a common platform. More importantly, the optoelectronic, spin and carrier transport properties of lead-based hybrid materials are also known to exhibit exceptional defect tolerance [Akkerman et al, Nature Materials, 17 394-405, 2018]. Formation of shallow trap states, when these materials are processed using cruder fabrication methods than conventional electronic materials, contributes to their fault tolerance, which is a crucial material design requirement for spin-based applications at room temperature. This suggests that hybrid materials in some aspects are indeed superior to the conventional semiconductor materials that require ultra-high purity composition using sophisticated high-temperature deposition approaches since their properties are extremely susceptible to the defects (i.e., very low defect tolerance).

At the current stage, the spin propagation length proposed in this work is around 600 nm that is somehow lower than that in Si or Graphene as the second referee pointed out, yet hybrid perovskite materials exhibit a vast and unexplored chemical "universe" which includes perovskite, Ruddlesden-Popper (used in this work), and Dion-Jacobson structures, combining various metal halides and organic molecular cations. Thus the work in this manuscript would launch a promising route of designing a wide variety of reduced-dimensional hybrid perovskites materials for future solution-based spintronic and spin-optoelectronic applications, whereas the similar materials development in the class of conventional semiconductors is limited.

In the current version of the manuscript, the authors have significantly improved the presentation, rephrased the discussion, and fully addressed all the previous comments raised by both referees. Therefore, I recommend the publication of this manuscript in Nature Communications as it is.

Reviewer #3 (Remarks to the Author):

Comments,

In this paper, the authors demonstrated that the room-temperature long-range spin transport via spin-funneling effect in Ruddlesden-Popper perovskites (RPP). Efficient ultrafast funneling of photoexcited spin-polarized excitons can be seen at room temperature and this interesting phenomenon can be interpreted in terms of the effect of the intermediate states-assisted energy transfer process during the thermalization. The authors also claimed that the use of the graded RPP film can achieve the uni-directional out-of-plane spin-funneling via 600 ± 80 nm, which is comparable transfer of spin information as shown in organic and inorganic semiconductors.

Considering the authors' claims and presentations in the manuscript and response letter, I think that there are some unfair comparisons to increase the impact of the physics presented in this study. Recent studies on graphene (e.g., 2D Materials 5, 032004 (2018)) and Si (e.g., Phys. Rev. Applied 2, 034005 (2014) & Phys. Rev. B 95, 115302 (2017).) achieved the spin transport at room temperature over a few microns (not a few hundreds of nm). It seems that the authors intentionally selected some convenient works as references to increase the impact of the physics presented here. In addition, because such previous works on graphene and Si have already used

ELECTRICAL spin injection and detection through the spin transport (not optical methods), the previous works are more suitable for the device applications in the field of spintronics. Although the authors unfairly described in INTRODUCTION that "Significant research efforts are devoted to search for a suitable candidate, but to no avail.", these statements are not appropriate for readers in the field of semiconductor spintronics. At least, the spin transport at room temperature has already been reported in graphene, GaAs, Si, Ge, GaN, ... by the useful electrical methods. On the other hand, the present study only shows an optical spin generation, which cannot be used for spintronics applications. Therefore, the authors should largely improve the statements on the comparison between this study and other semiconductor spintronics at room temperature in INTRODUCTION and DISCUSSION. Furthermore, in this manuscript, I cannot see any data on Hanle effect or magnetic-field dependence. In general, the decay of the generated spin polarization by applying magnetic fields should be given to judge the validity of the experimental data. In previous reviewers' report, this point was described. In Refs. 13 and 14, the spin polarization and/or spin related signals as a function of magnetic field have clearly been shown as evidences for the relaxation of the generated spin polarization by applying external magnetic fields perpendicular to the polarized spins. For publication in Nature Communications, the authors should add some data on the decay of the spin polarization by applying the perpendicular magnetic fields, in particular, in Fig. 4 c. Although the spin-funneling effect in RPP films is interesting, there is a gap in the authors' logic to understand the impact of the physics presented in this study. Finally, this manuscript included lots of English and typographical errors.

Reviewer #4 (Remarks to the Author):

The paper presents unexpected ultra fast spin polarization of excitons in mixed phase Ruddlesden Popper halide perovskite at room temperature. Authors ascribe the corresponding relaxation pathway to the presence of quantum coupled intermediate states that contribute to the so called "spin funneling" effect involving different material phases. This hypothesis is supported by Transient Absorption spectroscopy with circularly polarized pump and probe. The paper is supported by a rich section discussing additional data and models.

The paper is well written and the reader can fluently follow the discussion. There are nevertheless some unclear points that need to be addressed to make the paper ready for publication. I proceed by commenting directly the figures.

Figure 2:

Fig2d presents two different trends in the spin polarization of mixed phase sample ($n_{\text{bar}}=4$) indicating almost an increase of 50% of polarization close to 0.2 eV at ΔE ($E_{\text{pump}} - E_{\text{probe}}$), followed by a rapid drop, a sort of plateau and again an increase at higher thermalization energy. Authors claim that such finding is a prove of sequential transfer of lower n phases with higher band gap. But why should we expect such plateau in the intermediate states between $0.2\text{eV} < \Delta E < 0.4\text{eV}$? A more detailed discussion on qualitative trend should be included. Moreover, data presented in fig4c show almost an opposite behaviour (see comments below). I understand that samples are drastically different but a comparative analysis should be included for clarity.

Fig. 2e-f. For MAPI sample, bleaching data are presented by considering a fixed energy for the probe and variable energy for the pump while the opposite is for the mixed phase. Authors need to comment clearly the reasons for presenting such data in different manner or make the two presentation consistent.

Fig 3:

Fig 3c. Why the PA and PB signals do not converge at the same zero as in the $n_{\text{bar}}=4$ sample? What is the origin of such differences?

Fig. 3e At which sample it refers?-add details in the caption-

The fact that co-circular data does not saturate is really counterintuitive, authors ascribe it to a multi particle process for which the saturation is not attained. Why I should expect a linearly

increasing with fluence? What is the physical origin of such multi particle process? Any comparative result with other mixed phase compounds?

Fig4.

This part is the less clear to me, because the claim of long spatial distance travelled by spin information is not well explained.

I do not see huge difference between Fig4b(graded sample) and fig 3b($n_{\text{bar}}=4$ sample), nevertheless the Spin polarization deduced in fig 4c and fig 2d ($n_{\text{bar}}=4$ sample) present different trends (please use the same method to define X data for better comprehension of the data). The claim "This implies a better formation of $n \gg 1$ phases, in contrast with the standard $n_{\text{bar}} = 4$ sample" it is not clear in terms of advantages in the funneling. Such a sentence seems to correlate the results to the presence of single phases, but this is not the case. Are the grain boundaries providing such intermediate states? Any difference between thickness dependence of the graded sample and the $n_{\text{bar}}=4$ sample?

Is the unidirectionality provided by the deposition method? What is the real microstructure of the sample?

As general comment, I did not find in the paper a possible origin for this coupled intermediate states. From the data one can argue that grain boundaries play a major role. Other possible explanations?

In addition:

- Provide in the text the definition of ΔT . Even if trivial it helps to understand discussion at pag.6
- Check the acronym, es. TA is presented at pag 5 but defined at pag 6
- Last section is named discussion, while conclusion is more suitable

In summary, this paper could be considered for publication in Nat. Commun. once such minor comments will be addressed.

Response to the reviewers

We thank all the Reviewers for their comments, which have helped us further strengthen our manuscript. Appended in this letter, we have addressed to the best that we could all the Reviewers' concerns about the manuscript and have revised the manuscript accordingly. Herein, reviewers' comments are given in *green italics*, our responses are given in black, and our highlighted revisions in the manuscript are given in *blue italics*. The deleted parts are given in *brown italics*.

Reviewer #1

It is my second to review this manuscript (it was submitted to Nature Photonics). As I stated earlier in my previous comment: "The discovery of such a long-time spin-dependent exciton would potentially introduce a novel concept of the transport of spin information in solution-processed hybrid perovskite thin films for quantum computing applications and communication technologies." It is noteworthy that these unique low-temperature solution-processed hybrid materials encompass rich spin-dependent phenomena (e.g., giant Rashba splitting, spin-selective optical Stark effect, spin control induced by chirality, etc.) that are designed to manipulate within the same material class, and they can be synthesized using a common platform. More importantly, the optoelectronic, spin and carrier transport properties of lead-based hybrid materials are also known to exhibit exceptional defect tolerance [Akkerman et al, Nature Materials, 17 394-405, 2018]. Formation of shallow trap states, when these materials are processed using cruder fabrication methods than conventional electronic materials, contributes to their fault tolerance, which is a crucial material design requirement for spin-based applications at room temperature. This suggests that hybrid materials in some aspects are indeed superior to the conventional semiconductor materials that require ultra-high purity composition using sophisticated high-temperature deposition approaches since their properties are extremely susceptible to the defects (i.e., very low defect tolerance).

At the current stage, the spin propagation length proposed in this work is around 600 nm that is somehow lower than that in Si or Graphene as the second referee pointed out, yet hybrid perovskite materials exhibit a vast and unexplored chemical "universe" which includes perovskite, Ruddlesden-Popper (used in this work), and Dion-Jacobson structures, combining

various metal halides and organic molecular cations. Thus the work in this manuscript would launch a promising route of designing a wide variety of reduced-dimensional hybrid perovskites materials for future solution-based spintronic and spin-optoelectronic applications, whereas the similar materials development in the class of conventional semiconductors is limited.

In the current version of the manuscript, the authors have significantly improved the presentation, rephrased the discussion, and fully addressed all the previous comments raised by both referees. Therefore, I recommend the publication of this manuscript in Nature Communications as it is.

We are delighted by Reviewer #1 encouraging comments and constructive scientific feedback. We are greatly appreciative for all his/her inputs to further raise the standards of our manuscript to even higher levels.

Reviewer #3

In this paper, the authors demonstrated that the room-temperature long-range spin transport via spin-funneling effect in Ruddlesden-Popper perovskites (RPP). Efficient ultrafast funneling of photoexcited spin-polarized excitons can be seen at room temperature and this interesting phenomenon can be interpreted in terms of the effect of the intermediate states-assisted energy transfer process during the thermalization. The authors also claimed that the use of the graded RPP film can achieve the uni-directional out-of-plane spin-funneling via 600 ± 80 nm, which is comparable transfer of spin information as shown in organic and inorganic semiconductors.

Considering the authors' claims and presentations in the manuscript and response letter, I think that there are some unfair comparisons to increase the impact of the physics presented in this study. Recent studies on graphene (e.g., 2D Materials 5, 032004 (2018)) and Si (e.g., Phys. Rev. Applied 2, 034005 (2014) & Phys. Rev. B 95, 115302 (2017).) achieved the spin transport at room temperature over a few microns (not a few hundreds of nm). It seems that the authors intentionally selected some convenient works as references to increase the impact of the physics presented here.

We thank the Reviewer for the pointers. As we were not aware of those mentioned publications, we did not intentionally select the cited publications to increase our impact. In fact, we have also fairly highlighted the weakness of perovskites for spintronic applications: (Page 3, Line 17) “... *Nevertheless, this also restricts the spin lifetime to only a few ps in room temperature and up to ~1 ns in cryogenic temperature. This constraint would limit effective spin transport and could potentially derail halide perovskite’s spintronics aspirations*”. Nevertheless, as commented by the Reviewer, without mentioning the achievements of Graphene and Si, we might have given the erroneous impression of intentionally exaggerating the impact of our findings. Based on this feedback, we have added discussion on Graphene and Si and cited the mentioned publications. Therefore, we believe that this would be a fairer presentation of our findings and their impact into the spintronics community.

However, it is also pointed out by Reviewer #1 that by just merely comparing the spin propagation lengths, our work seems inferior to Si or Graphene, “*hybrid perovskite materials exhibit a vast and unexplored chemical “universe”.*” The purpose of our work is not to claim the discovery of the “best material for spintronics”, but to demonstrate the potential of hybrid perovskites as an alternative platform for such spin-related applications. This is a crucial step forward in any fledging new material system. We strongly believe that hybrid perovskite materials deserve a fair chance for development just like Graphene during its early formative years.

Moreover, while Graphene and Si have their advantages in terms of spin propagation lengths, perovskites also have their own merits. As mentioned in our response to the previous review, we recognized that there are also at least three limitations of the comparison, which could be overcome by perovskites. (1) Both Si and Graphene have very low spin-orbit coupling (SOC), in which, while long spin diffusion via spin-drift are granted, fundamentally limits the viability for spin-control (both optically and electrically). (2) These two systems would require good quality single-crystal samples with minimal scattering centres to achieve long-range spin-drift, thus requires costly fabrication techniques. (3) Lattice mismatch and unintentional doping due to substrate effect are also major problems in Si and Graphene, respectively.

Based on this feedback, we have made the following revisions into the manuscript:

- Page 3, Line 3: Short discussion on the spintronics progress in graphene and Si is added: *“Recently, spintronic studies on low SOC inorganic system, such as graphene¹⁰ and Si¹¹ have yields exciting results, where room temperature spin-transport length of ~30 μm and ~20 μm, respectively, have been demonstrated. However, they are also limited by several fundamental issues: (1) both Si and graphene have very low spin-orbit coupling (SOC), in which, while it grants long spin diffusion, fundamentally limits the viability for spin-control (optically and electrically). (2) These two systems would require good quality single-crystal samples with minimal scattering centers to achieve long range spin-drift, thus requires costly fabrication techniques. (3) Lattice mismatch and unintentional doping due to substrate effect are also major problems in Si and graphene, respectively.”*
- Page 3, Line 13: Short comment on the advantage of perovskites is added: *“The defect tolerance and solution-processability of lead halide perovskites might offer advantages over other material systems.”*
- Page 12, Line 23: The sentence *“This is the possible exciting outlook derived from our current result, which we are working towards.”* is changed into *“This is the possible exciting outlook derived from our current result, which paves the way for future research.”*

In addition, because such previous works on graphene and Si have already used ELECTRICAL spin injection and detection through the spin transport (not optical methods), the previous works are more suitable for the device applications in the field of spintronics. Although the authors unfairly described in INTRODUCTION that “Significant research efforts are devoted to search for a suitable candidate, but to no avail. ...”, these statements are not appropriate for readers in the field of semiconductor spintronics. At least, the spin transport at room temperature has already been reported in graphene, GaAs, Si, Ge, GaN ... by the useful electrical methods. On the other hand, the present study only shows an optical spin generation, which cannot be used for spintronics applications. Therefore, the authors should largely improve the statements on the comparison between this study and other semiconductor spintronics at room temperature in INTRODUCTION and DISCUSSION.

We are afraid that the Reviewer might have misunderstood our point. It is not our intention to unfairly downplay the achievements of Graphene and Si in electrical spintronics. However, we note that optical spin injection and detection is also an active research area on its own accord. For instance, some examples (with recent publications cited) are spintronic-based all-

optical technology [e.g. *Nat. Commun.* 10, 110 (2019); *Phys. Rev. Appl.* 11, 034001 (2019); *Appl. Sci.* 8(10), 1880 (2018); *Nat. Mater.* 13, 286–292 (2014)], spintronic metamaterials and plasmonics [e.g. *ACS Photonics* 5(10), 3956–3961 (2018); *Proc. SPIE 10919, Oxide-based Materials and Devices X, 109191C* (2019)], control of magnetic interactions [e.g. *Appl. Sci.* 9(5), 948 (2019)], quantum emitters [e.g. *Nat. Commun.* 7, 11820 (2016)], etc. Moreover, while our present work focuses on demonstrating optical spin generations, this novel concept could also be extended into electrical measurements in future work. In fact, it is fair to state that many of the spin studies in fledging new material systems actually began with optical studies. Nonetheless, we fully appreciate the Reviewer’s comments. We have improved and cautioned our earlier statements and comparisons of the current studies. Accordingly, we have made some amendments to the revised manuscript:

- Abstract, Line 2: The sentence “*Presently, the lack of suitable materials is the bottleneck of this technology.*” is changed to “*Presently, there are few suitable material systems*”.
- Page 2, Line 18: The sentence “*Significant research efforts are devoted to search for a suitable candidate, but to no avail.*” is changed to “*Significant research efforts are devoted to search for suitable candidates.*”
- Page 2, Line 19: The sentence “*Forerunner research on inorganic semiconductors (e.g., GaAs¹⁻⁴, MnSe⁵, Si⁶, etc.) face not only cost and integration challenges from a high vacuum and temperature processing and lattice-matching requirements but also, in most cases, limitation to cryogenic working temperature.*” is changed to “*Forerunner research on inorganic semiconductors (e.g., GaAs¹⁻⁴, MnSe⁵, Si⁶, etc.) face not only cost and integration challenges from a high vacuum and temperature processing and lattice-matching requirements.*”

Furthermore, in this manuscript, I cannot see any data on Hanle effect or magnetic-field dependence. In general, the decay of the generated spin polarization by applying magnetic fields should be given to judge the validity of the experimental data. In previous reviewers’ report, this point was described. In Refs. 13 and 14, the spin polarization and/or spin related signals as a function of magnetic field have clearly been shown as evidences for the relaxation of the generated spin polarization by applying external magnetic fields perpendicular to the polarized spins. For publication in Nature Communications, the authors should add some data on the decay of the spin polarization by applying the perpendicular magnetic fields, in particular, in Fig. 4c.

We thank the Reviewer for the comment. As mentioned in the previous review process, due to our currently limited experimental capability, we are unable to perform the optical or electrical Hanle measurement. Nonetheless, to address the Reviewer’s concern, we have designed and performed another experiment that captures the same essence as Hanle effect measurement, on our $\bar{n} = 4$ sample. Herein, we performed the measurement of exciton spin relaxation time with and without magnetic field perpendicular to the sample surface (Fig. R1a). Similar as the case of typical Hanle effect measurement, the presence of magnetic field perpendicular (at y -direction) to the initial photogenerated spin-polarization (along z -direction) triggers the spin precession along the y -axis, and therefore lowers the spin-polarization (or shortens the spin lifetime) detected optically in the z -direction.

Unfortunately, to the best of our capabilities, we could only perform such measurement at room temperature (~ 300 K) with maximum magnetic field (B-field) of ~ 0.3 T. Based on the parameters derived from Ref. 13 (Larmor frequency $\omega_L/2\pi$ of ~ 10 GHz at B-field of 300 mT), with a spin lifetime τ_s of ~ 2 ps, we expect a spin lifetime shortening due to B-field by a factor of $(\omega_L \tau_s)^2 / [1 + (\omega_L \tau_s)^2] \sim 2\%$. As expected, we indeed observed a small, yet consistent shortening of the measured spin lifetime – Fig. R1b. We regret that we could not make the experiment/result any clearer, as measurement at low temperature (at 10 K, as in Ref. 13 and 14) with a higher magnetic field is beyond our capabilities. Nevertheless, we believe that our current result is sufficient to validate our experimental data.

Figure R1 | Modified Hanle effect measurement in perovskites. (a) The circularly-polarized pump-probe geometry for measurements of the spin lifetime, in the presence of a perpendicular magnetic field. The spin-precession along y -axis induced by the B-field will shorten the lifetime of the detected spin-polarization at z -direction. (b) The exciton spin lifetime measured at different probe energies at $\bar{n} = 4$ RP perovskite sample. Small yet consistent shortenings of spin-lifetimes are observed in the presence of a B-field.

This discussion has been added into Section 11 in our revised Supplementary Information (SI).

- SI Section 11: This discussion on the “modified” Hanle effect measurement has been added.

Although the spin-funneling effect in RPP films is interesting, there is a gap in the authors' logic to understand the impact of the physics presented in this study.

We fully appreciate the Reviewer's opinion. As explained previously, we might have over-claimed our impact, as we were not fully aware of the publications that have been mentioned by the Reviewer. We have therefore revised our manuscript and cautioned our claims as discussed in the previous comment.

Finally, this manuscript included lots of English and typographical errors.

We thank the Reviewer for the feedback. We apologise if some of the typos had eluded us. The manuscript has now been proof-read many times by several colleagues, including native English speakers. We feel that this issue has now been resolved.

Reviewer #4

The paper presents unexpected ultrafast spin polarization of excitons in mixed phase Ruddlesden Popper halide perovskite at room temperature. Authors ascribe the corresponding relaxation pathway to the presence of quantum coupled intermediate states that contribute to the so called “spin funneling” effect involving different material phases. This hypothesis is supported by Transient Absorption spectroscopy with circularly polarized pump and probe. The paper is supported by a rich section discussing additional data and

models.

The paper is well written, and the reader can fluently follow the discussion. There are nevertheless some unclear points that need to be addressed to make the paper ready for publication. I proceed by commenting directly the figures.

We thank the Reviewer for the encouraging comments. Following are our answer to the detailed comments by the Reviewer:

- 1. Figure 2: Fig 2d presents two different trends in the spin polarization of mixed phase sample ($n_{\text{bar}} = 4$) indicating almost an increase of 50% of polarization close to 0.2 eV at ΔE ($E_{\text{pump}} - E_{\text{probe}}$), followed by a rapid drop, a sort of plateau and again an increase at higher thermalization energy. Authors claim that such finding is a prove of sequential transfer of lower n phases with higher band gap. But why should we expect such plateau in the intermediate states between $0.2 \text{ eV} < \Delta E < 0.4 \text{ eV}$? A more detailed discussion on qualitative trend should be included.*

We thank the Reviewer for the excellent question. The trend can be qualitatively explained as follows. The pump photoexcites resonantly the $n = 3$ phase and non-resonantly other $n > 3$ phases. The initial increase followed by a decrease at $\Delta E < 0.2 \text{ eV}$ arises from the interplay between the spin-funneling process from $n = 3$ to $n = 4$ and 5, and thermalization of excitons in $n = 4$ and 5 phases within itself. The plateau at $0.2 \text{ eV} < \Delta E < 0.4 \text{ eV}$ signifies that the spin polarization at this region originates dominantly from the spin-funneling. Meanwhile, the increase of the spin-polarization at the end of the plateau arises from density-dependent diffusion/funneling due to traps, as described in our trap-limited energy diffusion model. In fact, the trend of the observed spin-polarization in the standard sample could also be semi-quantitatively reproduced by our model with F -test confidence of 95% with respect to the data mean (*i.e.*, $F = 6.14$, with $p = 6$).

Based on Reviewer's feedback, we have revised the following parts of the manuscript to improve its readability:

- Fig. 2d, Main Text: has been revised from:

to the following:

- Figure caption (Fig. 2d): explanation of on the model fitting “... *function of $\Delta E = E_{pump} - E_{probe}$, with average spin-polarization 0.46 ± 0.05 . The spin polarization data in (d) (top) is fitted by our trap-limited energy diffusion model (see SI). ...*” is added.
- Page 6, Line 19: Qualitative explanation on the trend is added: “*An F-test on the model shows 95% significance ($F = 6.14$ with $p = 6$, against the mean), which confirms the validity of the model. Qualitatively, this trend can be described as the following. The initial increase followed by a decrease at $\Delta E < 0.2$ eV arises from the interplay between the spin-funneling process from $n = 3$ to $n = 4 - 5$ phases, and thermalization of exciton within $n = 4 - 5$ phases itself. It is then followed by a plateau at 0.2 eV $< \Delta E < 0.4$ eV, which signifies spin polarization that*

predominantly originates from the spin-funneling; and the final increase arises from density-dependent diffusion/funneling due to traps.”

Moreover, data presented in fig4c show almost an opposite behaviour (see comments below). I understand that samples are drastically different, but a comparative analysis should be included for clarity.

We thank the Reviewer for bringing up this issue. Here, the Reviewer might have misunderstood our data representation. The trends between the $\bar{n} = 4$ sample (Fig. 2d) and the 2D/3D graded sample (Fig. 4c) are actually similar, but only represented in two different x -axes. Fig. 2d was presented with $E_{pump} - E_{probe}$ as the x -axis, while Fig. 4c was presented with E_{probe} as the x -axis. As the result, the trends appeared “mirrored”. We apologize for this potentially confusing data representation and have revised the manuscript accordingly.

Based on this comment, we have made the following change(s) to the manuscript:

- Fig. 4c, Main Text: has been revised from:

unto the following:

- Figure caption (Fig. 4c): changed from “... (c) Initial photoexcited spin-polarization of our 2D/3D sample (green) as function of *probe energy*, with average spin-polarization 0.36 ± 0.03 ...” to “... (c) Initial photoexcited spin-polarization of our 2D/3D sample (green) as function of $\Delta E = E_{\text{pump}} - E_{\text{probe}}$, with average spin-polarization 0.36 ± 0.03 ...”

Fig. 2e-f. For MAPI sample, bleaching data are presented by considering a fixed energy for the probe and variable energy for the pump while the opposite is for the mixed phase. Authors need to comment clearly the reasons for presenting such data in different manner or make the two presentations consistent.

We thank the Reviewer for raising this comment. The rationale behind such an experimental design is as follows. For the MAPI sample in Fig. 2e-f, the photoexcited population thermalizes within $\sim 200\text{-}300$ fs (limited by our temporal resolution) from the pump energy level to the band-edge, with metastable states in between. The population thus manifests itself as photobleaching (PB) signal only at the band-edge (~ 1.6 eV). There is indeed also a broadband photoinduced absorption (PA) signal observed at probe energies between the pump level and the band-edge. However, this PA signal does not reflect the population of the metastable states, but rather due to other photophysical processes originating from the band-edge population. Hence, the observation of spin-polarized population could only be done at the band-edge PB signal, as it is a direct manifestation of the population. Therefore, we have to fix the probe energy and vary the pump energy to study the thermalization effect on the spin relaxation.

Meanwhile, for mixed-phase RP samples in Fig. 2c-d, there is a broad range of stable states (not meta-stable as in MAPI case), spanning from $n = 3$ band-edge at ~ 2.02 eV up to $n \rightarrow \infty$ band-edge at ~ 1.70 eV. These stable states arise from the band-edges of different RP phases in between. As a consequence, a broadband PB signal that reflects the populations of different RP phases at different probe energies is observed. Since we intend to study the effect of spin-funneling between these states, we have to vary the probe energy to monitor the spin-polarization of each of the n states. Moreover, to reduce the contribution from thermalization to the observed spin-polarization, we have to fix the pump energy near the resonance of $n = 3$ to minimize the effect of thermalization from the pump-level to the $n = 3$ band-edge.

2. *Fig 3: Fig 3c. Why the PA and PB signals do not converge at the same zero as in the $\bar{n} = 4$ sample? What is the origin of such differences?*

The reviewer asked an excellent question. Based on our interpretation, in this region between the resonances of $n = 3$ and $n = 4$ phases, the PB signal arises due to stimulated emission (SE) from the intermediate states (IS), while the PA band arises from the excited-state absorption from trapping within the IS. Following spin relaxation, the intensity of the signal is determined by the ratio between the SE transition strength and the PA transition strength from the traps. Thus, this signal strongly depends on the trap densities, type of samples, and preparation techniques. Nevertheless, it does not affect any of our interpretation.

In fact, we have explained it in previously in our manuscript: (Page 10, Line 6) “*The remaining weak TA signal would then originate from the difference in transition-strength between PA and SE from IS. This resultant signal is strongly dependent on the IS trap densities, which would differ based on sample types and preparations. It explains the difference of our observation of positive ($SE > PA$) and negative ($SE < PA$) post- spin relaxation TA signal for our $\bar{n} = 2$ and $\bar{n} = 4$ samples, respectively (Fig. 3c and d)*”. Thus, we hope that this explanation clarifies the Reviewer’s question.

Fig. 3e, at which sample it refers?-add details in the caption.

We thank the Reviewer for pointing it out. The data in Fig. 3e comes from $\bar{n} = 4$ standard sample. We have added this information to the figure caption.

- Figure caption (Fig. 3e): The detailed information “... counter-circular signal on $\bar{n} = 4$ sample at 1.98 eV probe, showing linear and ...” has been added.

The fact that co-circular data does not saturate is really counterintuitive, authors ascribe it to a multi particle process for which the saturation is not attained. Why I should expect a linearly increasing with fluence? What is the physical origin of such multi particle process? Any comparative result with other mixed phase compounds?

The Reviewer might have misunderstood our explanation. We mentioned in the manuscript that the linear increase of the co-circular signal implies that neither multi-particle processes nor pump saturation regime has been reached with our pump fluence, i.e. (Page 9, Line 2): “(iii) *The $\Delta T/T$ signal peak intensities for co- and counter-circular system show linear and saturating trends with increasing pump fluence, respectively –*

Fig. 3e. The former implies that no multi-particle process is involved and the saturation of pump absorption has not been reached; ...". Hence, since there is no multi-particle process involved, we can safely attribute the saturation of the counter-circular signal to the trap-filling.

- 3. Fig4: This part is the less clear to me, because the claim of long spatial distance travelled by spin information is not well explained. I do not see huge difference between Fig. 4b (graded sample) and Fig. 3b ($n_{\text{bar}} = 4$ sample), nevertheless the Spin polarization deduced in Fig. 4c and Fig. 2d ($n_{\text{bar}} = 4$ sample) present different trends (please use the same method to define X data for better comprehension of the data).*

We apologize for being unclear in our explanation. As mentioned in the previous comments, both graded and standard sample show a similar spin-polarization trend, yet presented with two different x-axes in the previous version of the manuscript. This matter has been addressed in the revised manuscript.

The claim "This implies a better formation of $n \gg 1$ phases, in contrast with the standard $n_{\text{bar}} = 4$ sample" it is not clear in terms of advantages in the funneling. Such a sentence seems to correlate the results to the presence of single phases, but this is not the case.

The Reviewer might have misunderstood our point. The sentence "*This implies a better formation of $n \gg 1$ phases ...*" was only used to explain the broader range of the photobleaching (PB) band in the TA signal of our 2D/3D graded samples. In Ruddlesden-Popper perovskites, the bandgap of each phase is well-defined in the energy domain. Therefore, the range of the PB band towards the pure 3D band-edge (~ 1.6 eV) will denote the density of the "bulk-like" or " $n \gg 1$ " phases formed in our samples.

Are the grain boundaries providing such intermediate states?

We thank the Reviewer for the suggestion. As of now, apart from spectroscopically demonstrating it, we could not yet pinpoint the exact origin of these intermediate states. There are several possibilities including edge states [*Science* 355 (6331), 1288 (2017)], inter-well two-exciton excited states [*J. Phys. Chem. Lett.* 8 (16), 3895 (2017)], tail states, or grain boundaries.

Based on Reviewer's suggestion, we have made the following amendment in the manuscript:

- Page 9, Line 19: Comments on the origin of the IS: "*These IS could originate from either edge states²⁷, inter-well two-exciton excited states²⁸, tail states, or grain boundaries. However, pinpointing the origin of these states is beyond the scope of current work.*" is added.

Any difference between thickness dependence of the graded sample and the $n_{\text{bar}} = 4$ sample?

We thank the Reviewer for the question. In the graded sample, the spin-polarized excitons have to travel across the sample thickness in order to funnel to low bandgap phases. Hence, a thickness dependence is present, as shown in Fig. 4d. On contrary, for the standard samples, the distribution of the RP phases is randomized. Therefore, there should not be any correlation between the distance travelled by the spin-polarized excitons and the sample thickness. This comparison is analogous to the case of charge transport from active materials between a planar and a mesoscopic photovoltaic cell.

Is the unidirectionality provided by the deposition method? What is the real microstructure of the sample?

The Reviewer asked an excellent question. Yes, the uni-directionality of the graded samples is provided by the addition of DMSO solvent and $\text{CH}_3\text{NH}_3\text{Cl}$ additive in the precursor solution (in anhydrous N,N-Dimethylformamide, *i.e.*, DMF). The graded structure emerges from the interplay of different solubilities of the 2D and 3D perovskite precursor ions, and different boiling points of the solvent mixture. This synthetic procedure is described in detail in a previous publication [*Adv. Energy Mater.* 8, 1800185 (2018)]. The best illustration of the graded sample microstructure is presented in Fig. 4a. Different reflection and PL spectra when the samples were photoexcited from front or back (substrate) sides, also confirms the graded structure – SI Section 10. Unfortunately, a direct measurement/imaging of the sample microstructure across its cross-section is beyond our current capability.

As general comment, I did not find in the paper a possible origin for this coupled intermediate states. From the data one can argue that grain boundaries play a major role. Other possible explanations?

We thank the Reviewer for the comment. This matter has been addressed in Reviewer's previous comment.

In addition:

- *Provide in the text the definition of ΔT . Even if trivial it helps to understand discussion at page 6*

We thank the Reviewer for the feedback. We have added the following explanation to the Main Text based on the suggestion:

- Page 5, Line 8: Simple explanation on $\Delta T/T$ signal: “*Our hypothesis is proven by circular-polarized transient absorption (TA) spectroscopy (or known as pump-probe, see Methods for details). Herein, the change of probe transmission ($\Delta T/T$) induced by pump pulse could be used to measure the population of the exciton spin states, hence allows a time-resolve monitoring of the spin dynamics.*” has been added.

- *Check the acronym, es. TA is presented at page 5 but defined at page 6*

We thank the Reviewer for the feedback. This issue has been addressed in the highlighted revision of the previous comment.

- *Last section is named discussion, while conclusion is more suitable*

We thank the Reviewer for the feedback. We have renamed the headings from “*Results*” to “*Results and Discussions*”, and from “*Discussions*” to “*Conclusion*”, as suggested.

In summary, this paper could be considered for publication in Nat. Communic. once such minor comments will be addressed.

We are really grateful for all the constructive comments and feedback from the Reviewer, which has helped us strengthened our manuscript further.

REVIEWERS' COMMENTS:

Reviewer #3 (Remarks to the Author):

In the revised manuscript and response letter, the authors have politely amended all of the points in the previous comments raised by the reviewers. I also understood that the addition of the experimental data in Fig. S9 is the best effort at this stage. On the whole, since I am satisfied with the authors claims and improvements, I recommend the revised manuscript for publication in Nature Communications.

Reviewer #4 (Remarks to the Author):

Authors provided a detailed response to my concerns.
At this stage, I recommend publication.

Response Letter to the Reviewers

We thank the Reviewers for their time to provide valuable feedback to our manuscript. Herein, Reviewer's comments are highlighted in *blue italics* and our response is in black.

Reviewer #3:

In the revised manuscript and response letter, the authors have politely amended all of the points in the previous comments raised by the reviewers. I also understood that the addition of the experimental data in Fig. S9 is the best effort at this stage. On the whole, since I am satisfied with the authors claims and improvements, I recommend the revised manuscript for publication in Nature Communications.

We thank Reviewer #3 for his/her constructive comments and suggestions, which provided us with a broader perspective on spintronics research in Silicon and Graphene. We also thank the Reviewer for encouraging us to design the necessary experiments to further confirm our results and improve our manuscript.

Reviewer #4:

Authors provided a detailed response to my concerns. At this stage, I recommend publication.

We thank Reviewer #4 for his/her constructive comments and suggestions. These comments helped us to clarify our explanation on the photophysics, and therefore improve the readability of our manuscript to a broader range of readers.